# Better Detection of Peripheral Arterial Disease with Toe-Brachial Index Compared to Ankle-Brachial Index among Taiwanese Patients with Diabetic Kidney Disease

**DOI:** 10.3390/jcm12237393

**Published:** 2023-11-29

**Authors:** Chia-Wei Chang, Ya-Wen Sung, Yu-Ting Huang, Yong-Chuan Chung, Mei-Yueh Lee

**Affiliations:** 1School of Medicine, Kaohsiung Medical University, Kaohsiung 807, Taiwan; ccw525689@gmail.com; 2Department of Nursing, Kaohsiung Medical University Hospital, Kaohsiung 807, Taiwan; 860161.kmuh@gmail.com; 3Statistical Analysis Laboratory, Department of Medical Research, Kaohsiung Medical University Hospital, Kaohsiung Medical University, Kaohsiung 807, Taiwan; stakmuh@gmail.com; 4Department of Business Management, National Sun Yat-Sen University, Kaohsiung 803, Taiwan; terrychung0623@gmail.com; 5Administration Management Center, Kaohsiung Siaogang Municipal Hospital, Kaohsiung 812, Taiwan; 6School of Medicine, College of Medicine, Kaohsiung Medical University, Kaohsiung 807, Taiwan; 7Division of Endocrinology and Metabolism, Department of Internal Medicine, Kaohsiung Medical University Hospital, Kaohsiung 807, Taiwan; 8Department of Internal Medicine, Kaohsiung Medical University Gangshan Hospital, Kaohsiung 820, Taiwan

**Keywords:** peripheral artery disease, toe-branchial index, ankle-branchial index, cardio-ankle vascular index, diabetic kidney disease

## Abstract

Background: Ankle-brachial index (ABI) is a simple method for diagnosing peripheral artery disease (PAD) but has limited reliability in patients with diabetes mellitus (DM) because of medial artery calcification. Our study aims to investigate whether the toe brachial index (TBI) or the cardio-ankle vascular index (CAVI) has a better detection over the ABI for diagnosing PAD in diabetic kidney disease (DKD). Materials and Methods: A cohort of 368 patients (mean age 68.59 ± 13.14 years, 190 males and 178 females) with type 2 DM underwent ABI, TBI, and CAVI measurements at our outpatient clinic. Results: Of all enrolled patients, the TBI is significant in evaluating PAD, especially in patients whose chronic kidney disease (CKD) stage 3a with adjusted odds ratio (AOR) = 6.50, 95% confidence interval (CI) 1.63–25.97, *p* = 0.0080, stage 3b AOR = 7.47, 95% CI 1.52–36.81, *p* = 0.0135, and stage 4–5 AOR = 20.13, 95% CI 1.34–94.24, *p* = 0.0116. CAVI is also significant in CKD stage 1 with AOR = 0.16, 95% CI 0.03–0.77, *p* = 0.0223, stage 2 with AOR = 0.18, 95% CI 0.04–0.74, *p* = 0.0180, and stage 3a AOR = 0.31, 95% CI 0.10–0.93, *p* = 0.0375. Conclusion: TBI has a better yield of detection of PAD compared to ABI among Taiwanese patients with DKD. CAVI may play a role in the early stage of DKD.

## 1. Introduction

In patients with type 2 diabetes, cardiovascular disease (CVD) is the most common cause of mortality and morbidity [1]. Diabetes mellitus increases microvascular complications, which include retinopathy, nephropathy, and neuropathy. Also, it may lead to macrovascular complications. Furthermore, patients with CKD are related to elevated cardiovascular risk manifesting as CVD [2]. Based on a previous systemic review that assessed community-based studies for global prevalence and risk factors of PAD, DM ranked as the second major risk [3]. According to the National Health and Nutrition Examination Survey, DM and smoking were also the most significant risk factors for PAD [4].

The ABI is a simple and noninvasive method to assess PAD [5], and an ABI value < 0.9 is sensitive for the diagnosis of PAD. However, the application of this index to diabetic patients is considered questionable because of medial artery calcification, which is a nonobstructive calcification of the tunica media that occurs commonly in the arteries of older adults and diabetic kidney disease patients; these factors could mistakenly elevate the ABI value [6,7]. Therefore, considering the medial artery calcification, the TBI is an alternative measure to the ABI to evaluate for PAD [8], TBI may be more specific to diagnosing PAD and identifying patients at risk for CVD. Arterial stiffness is the lack of viscoelastic properties of the arterial wall due to many reasons, including DM, vascular calcification, and hypertension [9]. CAVI is a newly developed method used to assess arterial stiffness such as common iliac, femoral, and tibial artery levels. Arterial stiffness is not affected by blood pressure; therefore, high CAVI can also be a predictor of CVD in CKD patients. However, using CAVI as a tool to assess and evaluate PAD in DKD patients is still under investigation. Knowing that PAD, a disease of the major arteries caused by atherosclerosis, is a vascular complication of DM [10], our study aims to determine which of the three measures, ABI, TBI, and CAVI, has the highest detection of PAD among patients with diabetic kidney disease.

## 2. Materials and Methods

### 2.1. Study Patients

Patients with type 2 DM who visited the Endocrinology and Metabolism outpatient department of the Southern Taiwan Medical Center between February 2022 and May 2022 were included in the study. We excluded patients with type 1 DM (defined as a presentation with diabetic ketoacidosis, acute hyperglycemia symptoms with heavy ketonuria (≥3), or the continuous requirement of insulin in the year succeeding diagnosis). Finally, 368 patients (mean age 68.59 ± 13.14 years, 190 males and 178 females) were included in this study.

### 2.2. Ethics Statement

The study protocol was approved by the institutional review board of the Kaohsiung Medical University Hospital (KMUHIRB-E(I)-20210313). Informed consent was obtained in written form from patients and all clinical investigation was conducted according to the principles expressed in the Declaration of Helsinki. The patients gave consent for the publication of the clinical details.

### 2.3. Assessment of ABI, TBI, and CAVI

The values of ABI, TBI, and CAVI were determined from the measurements by VS-2000 (Fukuda Denshi Corporation., Tokyo, Japan), which automatically and simultaneously measured blood pressures and pulse volumes in both arms and ankles or toe using an oscillometric method [11,12,13]. ABI and TBI were measured after allowing the patient to rest in a supine position for at least 5 min. The ABI and TBI were calculated by the ratio of the ankle or toe systolic blood pressure divided by the arm systolic blood pressure. The ABI and TBI measurements were conducted once in each patient. PAD can be diagnosed noninvasively by segmental blood pressure measurement and calculating an ABI or TBI. The diagnosis of PAD was defined as ABI < 0.9 or ≥1.3 in either leg or TBI below 0.65. The validation of this automatic device and its reproducibility had been previously published [12].

CAVI integrates information about the elasticity of blood vessels and represents a novel parameter of vascular stiffness that does not depend on BP [14]. The mathematical expression to calculate CAVI values is described elsewhere [14,15] and is mainly based on substituting the stiffness parameters β and PWV in the following equation: β = 2ρ × 1/(SBP − DBP) × ln (SBP/DBP) × PWV^2^, where ρ is the blood density and PWV is the heart–ankle PWV (haPWV) measured between the origin of the aorta and the tibial artery at the ankle. The mean coefficient of variation of CAVI is <5%, which is small enough for clinical use. The CAVI values, according to age and gender, are classified as normal (CAVI < 8), borderline (8 ≤ CAVI < 9), and abnormal (CAVI ≥ 9) [15].

The above measurements of ABI, TBI, and CAVI were all performed by our single technician to ensure the standardization of the readings. 

### 2.4. Collection of Demographic, Medical, and Laboratory Data

Demographic and medical data including age, gender, and co-morbid conditions were obtained from medical records and interviews with patients. The cardiovascular diseases included histories of old myocardial infarction and stroke, ischemic heart disease, atherosclerotic cardiovascular disease, and angina. The body mass index (BMI) was calculated as the ratio of weight in kilograms divided by the square of height in meters. Laboratory data were measured from fasting blood samples using an autoanalyzer (Roche Diagnostics GmbH, D-68298 Mannheim COBAS Integra 400, Basel, Switzerland). Serum creatinine was measured by the compensated Jaffé (kinetic alkaline picrate) method in a Roche/Integra 400 Analyzer (Roche Diagnostics, Mannheim, Germany) using a calibrator traceable to isotope-dilution mass spectrometry [16]. The value of eGFR was calculated using the 4-variable equation in the Modification of Diet in Renal Disease (MDRD) study, and CKD staging was as follows: stage 1 for estimated glomerular filtration rate (eGFR) of >90 mL/min/1.73 m^2^, stage 2 for eGFR of 60–89 mL/min/1.73 m^2^, stage 3a for eGFR of 45–59 mL/min/1.73 m^2^, stage 3b for eGFR of 30–44 mL/min/1.73 m^2^, stage 4 for eGFR of 15–29 mL/min/1.73 m^2^, and stage 5 for eGFR of <15 mL/min/1.73 m^2^ [17]. Urine albumin and creatinine were measured on a spot urine sample by an autoanalyzer (COBAS Integra 400 plus; Roche Diagnostics, North America). The urine albumin–creatinine ratio (UACR) was categorized into three groups: normoalbuminuria with urine ACR 0–29 mg/gm, microalbuminuria with urine ACR 30–299 mg/gm, and macroalbuminuria with urine ACR > 300 mg/gm. Blood samples were obtained within 1 month of ABI measurement. In addition, information regarding patient medications, including angiotensin-converting enzyme inhibitors (ACEIs), angiotensin II receptor blockers (ARBs), β-blockers, calcium channel blockers, diuretics, HMG-CoA reductase inhibitors (statins) and fibrates, oral antidiabetic agents like sulfonylureas, metformin, meglitinides, thiazolidinediones, alpha-glucosidase inhibitors, dipeptidyl peptidase-4 inhibitor (DPP4-I), sodium-glucose co-transporter 2 inhibitors (SGLT2-I), and insulin, during the study period was obtained from medical records.

### 2.5. Statistical Analysis

Statistical analysis was performed using SPSS 19.0 for Windows (SPSS Inc. Chicago, IL, USA). The sample size was calculated using the G-power of binary logistic regression. The G-power of ABI is 66.54%, TBI is 99.99%, and CAVI is 98.92%. Data are expressed as percentages, mean ± standard deviation, or median (25th–75th percentile) for age, DM duration years, total cholesterol, triglyceride, low and high-density lipoproteins, fasting glucose, HbA1c, creatinine, BMI, and systolic and diastolic blood pressures. The differences between groups were checked using a Chi-square test for categorical variables and an independent t-test for continuous variables. Multiple forward logistic regression analysis after adjustment of age, sex, DM duration coronary artery disease, systolic and diastolic blood pressures, BMI, hemoglobin A1c, fasting glucose, triglyceride, total cholesterol, high-density lipoprotein (HDL)-cholesterol, low-density lipoprotein (LDL)-cholesterol, eGFR, microalbuminuria and CKD stages, anti-hypertensive and anti-diabetic medications, and lipid-lowering medication use were used to identify the factors associated with an abnormal ABI, TBI, and CAVI. A difference was considered significant if the *p*-value was less than 0.05.

## 3. Results

A total of 368 patients with type 2 DM were included (mean age 68.59 ± 13.14 years; 190 males and 178 females), with some having different stages of chronic kidney disease (CKD). 

Comparisons of baseline characteristics between the patients with (n = 336) and without (n = 32) a normal ABI are shown in Table 1A.

CKD is categorized into stages 1 to 5. The prevalence of a normal ABI (ABI > 0.9) is significantly higher than that of an abnormal ABI (ABI < 0.9), with a *p*-value of 0.0067. CKD stages 1, 2, and 3a had a higher prevalence in normal than abnormal ABI, (N = 36 (10.7%) versus N = 3 (9.4%), N = 56 (16.7%) versus N = 4 (12.5%), N = 87 (25.9%) versus N = 5 patients (15.6%), *p*-value = 0.0067). However, stages 3b and 4 + 5 had a higher prevalence of abnormal than normal ABI (N = 51 (15.2%) versus N = 9 patients (28.1%) and N = 27 patients (8.0%) versus N = 8 patients (25%), *p*-value = 0.067), respectively. Urine ACR < 30 mg/g and 30–299 mg/g had a higher prevalence in normal than in abnormal ABI (N = 149 (44.3%) versus N = 10 (31.3%) and N = 127 (37.8%) versus N = 10 (31.3%), *p*-value 0.0396). However, urine ACR > 300 mg/g had a higher prevalence in abnormal than in abnormal ABI (N = 60 (17.9%) versus N = 12 (37.5%), *p*-value = 0.0396). Patients with abnormal ABI had a higher systolic blood pressure (SBP) of 137.20 ± 16.92 mmHg than patients with normal ABI with an SBP of 131.50 ± 15.28 mmHg, with a *p*-value of 0.0487. Patients with abnormal ABI, N = 14 (43.8%), had a higher prevalence of cardiovascular disease than patients with normal ABI, N = 74 (22.0%), with a *p*-value of 0.0059. Metformin users had a higher prevalence in normal ABI, N = 241 (71.7%), than in abnormal ABI, N = 16 (50.0%), with a *p*-value of 0.0105.

Comparisons of baseline characteristics between the patients with (N = 290) and without (N = 78) a normal TBI are shown in Table 1B.

CKD was categorized into stages 1 to 5. CKD stage 3a, 3b, and 4 + 5 had a higher prevalence in abnormal TBI (TBI < 0.65) than normal TBI (TBI > 0.65) with N = 67 (23.1%) versus N = 25 patients (32.1%), N = 44 (15.2%) versus N = 16 patients (20.5%), and N = 22 (7.6%) versus N = 13 patients (16.7%), respectively, with a *p*-value of 0.0039. While CKD stage 1 and 2 had a higher prevalence in normal than abnormal TBI (N = 33 (11.4%) versus N = 6 (7.7%) and N = 49 (16.9%) versus N = 11 (14.1%), *p*-value = 0.0039), respectively. Patients with abnormal TBI had a higher prevalence of cardiovascular disease than patients with normal TBI (N = 31 (39.7%) versus N = 57 (19.7%), *p*-value = 0.0002). Meglitinide users had a higher prevalence in abnormal TBI than normal TBI (N = 8 (10.3%) versus N = 7 (2.4%), *p*-value = 0.0019). However, diuretics users had a higher prevalence in normal than in abnormal TBI (N = 22 (7.6%) versus N = 18 patients (23.1%), *p*-value = 0.0001).

Comparisons of baseline characteristics between the patients with (n = 200) and without (n = 168) a normal CAVI are shown in Table 1C. Patients with abnormal CAVI were older and had a longer DM duration than patients with normal CAVI 71.91 ± 8.64 versus 63.18 ± 11.16, with a *p*-value of 0.0001, and 17.77 ± 9.31 versus 14.13 ± 9.25, with a *p*-value of 0.0003, respectively. Patients with normal CAVI (28.06 ± 5.70) had a higher body mass index (BMI) than patients with abnormal CAVI (25.69 ± 5.20) with a *p*-value of <0.0001. Patients who used meglitinides and alpha-glucosidase inhibitors had a higher prevalence of abnormal CAVI (N = 12 (7.1%) versus N = 3 (1.5%), *p*-value = 0.0064, and N = 20 (11.9%) versus N = 11 (5.5%), *p*-value = 0.0276, respectively). Alpha-blocker users had a higher prevalence in normal CAVI than in abnormal CAVI (N = 16 (8.0%) versus N = 5 patients (3.0%), with a *p*-value of 0.0374]).

Table 2 shows the determinants of ABI, TBI, and CAVI in the study patients as determined in multivariable logistic regression analysis. CKD stages 3a, 3b, and 4 + 5 were strongly associated with TBI, with an adjusted odds ratio (AOR) of 6.5, 95% confidence interval (CI): 1.63–25.97, *p* = 0.0080; AOR 7.47, 95% CI: 1.52–36.81, *p* = 0.0135; and AOR 20.13, 95% CI: 1.96–206.92, *p*= 0.0116, respectively. The AOR of TBI shows a value consistently above 1.0 beginning at CKD 3a, implying that this measure is associated with the true CKD stages. Interestingly, CKD stages 1, 2, and 3a were strongly associated with CAVI (AOR 0.16, 95% CI: 0.03–0.77, *p* = 0.0223; AOR 0.18, 95% CI: 0.04–0.74, *p* = 0.0180; and AOR 0.31, 95% CI: 0.10–0.93, *p* = 0.0375, respectively). Older age and longer DM duration were strongly associated with the CAVI level (AOR 1.11, 95% CI: 1.00–1.02, *p* < 0.0001 and AOR 1.05, 95% CI: 1.01–1.09, *p* = 0.0282, respectively). 

Increased triglycerides and low-density lipoprotein levels were associated with CAVI (AOR 1.01, 95% CI: 1.07–1.16, *p* = 0.03120 and AOR 1.06, 95% CI: 1.00–1.12, *p* = 0.0388, respectively). Lower BMI was strongly associated with CAVI (AOR 0.92, 95% CI: 0.85–0.99, *p* = 0.0223). Higher SBP was strongly associated with ABI (AOR 1.12, 95% CI: 1.03–1.21, *p* = 0.0090). CVD was associated with both TBI and ABI (AOR 4.0, 95% CI: 1.76–9.11, *p* = 0.0010 and AOR 7.19, 95% CI: 1.75–29.59, *p* = 0.0063, respectively). Meglitinide users were associated with both TBI and CAVI (AOR 6.13, 95% CI: 1.05–35.87, *p* = 0.0444 and AOR 47.27, 95% CI: 2.63–848.93, *p* = 0.0089, respectively). DPP4 inhibitor users were associated with ABI (AOR 0.10, 95% CI: 0.01–0.67, *p* = 0.0177). Diuretic users were strongly associated with TBI (AOR 4.19, 95% CI: 1.36–12.90, *p* = 0.0125). Alpha-blocker was associated with CAVI (AOR 0.13, 95% CI: 0.03–0.60, *p* = 0.0088).

In summary, our results show that the TBI is significant in evaluating PAD, especially in patients with chronic kidney disease (CKD) stage 3a with adjusted odds ratio (AOR) = 6.50, 95% confidence interval (CI) 1.63–25.97, *p*= 0.0080, stage 3b AOR = 7.47 95% CI 1.52–36.81, *p* = 0.0135, and stage 4–5 AOR = 20.13, 95% CI 1.34–94.24, *p* = 0.0116. CAVI is also significant in CKD stage 1 with AOR = 0.16, 95% CI 0.03–0.77, *p* = 0.0223, stage 2 with AOR = 0.18, 95% CI 0.04–0.74, *p* = 0.0180, and stage 3a AOR = 0.31, 95% CI 0.10–0.93, *p* = 0.0375. Therefore, TBI has a better yield of detection of PAD compared to ABI with DKD.

## 4. Discussion

In our study, the first important finding is that TBI was strongly associated with CKD stages 3 to 5. Considering the medial artery calcification of the type 2 diabetes population with chronic kidney disease, the TBI is an alternative measure to the ABI to evaluate for PAD [8]. Another important finding of our study is that CAVI was associated with early stages of CKD, stages 1, 2, and 3a, in type 2 diabetes mellitus, which is a good tool to evaluate for arterial stiffness that can assess vascular wall stiffness in the aorta, femoral artery, and tibial artery [14,18]. 

With respect to PAD, several studies have demonstrated that patients with CKD have a high risk of developing PAD [19]. In addition, results from the United States (US) National Health and Nutrition Examination Survey (NHANES) have reported that the prevalence of PAD in patients with diabetes was higher than in those without diabetes, furthermore, the risk of PAD increased by 2.5 times in patients with eGFR < 60 as compared to eGFR ≥ 60 mL/min/1.73m^2^, especially with higher ratios in CKD stage 3 to 5 patients. [4]. In an observational study enrolling more than 400.000 patients who were referred to the Manitoba Centre for Health Policy in Manitoba Canada, it was found that PAD is more common in patients with eGFR < 60 mL/min/1.73m^2^ compared with those with eGFR ≥ 60 mL/min/1.73m^2^ and frequently leads to lower-limb complications [20]. In 2012, the Kidney Disease Improving Global Outcome (KDIGO) guidelines recommended that adults with non-dialysis CKD be regularly examined for signs of PAD and be considered as candidates for the prescription of evidence-based therapies (Grade 1B) [21]. This meant the new study should assess the burden of PAD in pre-dialysis CKD patients. In hospital-based studies, the prevalence of PAD is 2–3 times higher in patients with versus without T2DM [22]. Various methods, including ABI, TBI, and arterial Doppler, are available to diagnose PAD.

In America, using a TBI cut-off of 0.7, a study has detected PAD (angiographically proven) in dialysis patients characterized by an ABI > 0.9 [23]. In addition, one study in Europe evaluated the predictive impact of the TBI test in people with CKD and ESRD to show that PAD confirmed by a low TBI value was associated with increased all-cause mortality [24]. Furthermore, previous reports have demonstrated that varying degrees of medial arterial calcification are common in diabetic kidney disease patients, so TBI is recommended for diabetic patients and patients with CKD because medial arterial calcification is less common in the toe than in the ankle. For patients with CVD, because of medial arterial calcification in diabetic kidney disease patients, these factors could falsely elevate the ABI value [6,7]; therefore, TBI was alternatively the better tool to evaluate PAD in these patients.

Previous studies also indicated that an increase in the CAVI is closely associated with a decreased eGFR or increased albuminuria [25]. The present study found that patients with macroalbuminuria were associated with abnormal CAVI, whose UACR was over 300 mg/g, so this could be useful for diagnosing patients with diabetic kidney diseases.

The presence of metabolic syndrome has been related to high CAVI as a factor of arterial stiffness [26]. The risk factors of metabolic syndrome include central obesity, elevated blood pressure, high triglycerides, low high-density lipoprotein (HDL), and high fasting glucose [27]. In our study, SBP was associated with ABI. Previous studies have shown that the assessment of 4-limb SBP heterogeneity is useful in the identification of a high-risk group of PAD and/or increased left ventricular mass index (LVMI), irrespective of the presence of overt PAD, which meant ABI was related to SBP [28].

There are also some parameters, such as BMI, triglycerides, and LDL levels, that are strongly associated with CAVI in our study. A previous study proved that high triglycerides were strongly associated with high CAVI independent of multiple cardiometabolic risk factors. This can be explained by the statistics that show that the triglycerides were positively associated with CAVI. There was also a report showing that TG was found to be associated with a risk of higher CAVI [29]. In our study, triglycerides had a significant association with CAVI: the higher the triglyceride level, the higher the CAVI.

There is a linear association between CAVI and age [30]; as patients age, their CAVI levels tend to increase. The duration of DM is associated with CAVI; based on the review, the CAVI has a linear association with the duration of diabetes, so the CAVI level might be a suitable tool for evaluating the duration of diabetes: the longer the patients had DM, the more possible abnormal CAVI level may happen.

Alpha-blocker, which mainly acts on α-adrenergic receptors to produce vasodilation, reduces systemic vascular resistance, and achieves an antihypertensive effect, has the function of dilating blood vessels and lowering blood pressure; however, the CAVI is a parameter of vascular stiffness that does not depend on blood pressure [14], so it is a good tool to evaluate patients’ conditions. 

Diuretics are mainly used in renal disease to facilitate extracellular fluid volume control, reduce the chance of developing hyperkalemia, and lower blood pressure. Furthermore, based on the results of the Antihypertensive and Lipid-Lowering Treatment To Prevent Heart Attack Trial (ALLHAT) and Joint National Committee (JNC) 7, diuretics are recommended as preferred agents in the general population with hypertension to lower their blood pressure and reduce CVD risk [31]. Accordingly, of these chronic kidney disease patients who used diuretics, most of them had CVD risk. In our study, it was associated with TBI.

In our study, patients who used non-sulfonylureas meglitinides were associated with CAVI and TBI. When patients use sulfonylureas as medicine to control their type 2 DM, the risk of hypoglycemia needs to be considered, mainly because of the decreased clearance of insulin and oral hypoglycemic drugs and impaired renal gluconeogenesis [32]. The patients with advanced stages of CKD 3 to 5 should consider short-acting non-sulfonylureas meglitinides as their priority to avoid hypoglycemia events, this might explain why meglitinide use was associated with the TBI in CKD stages 3 to 5 and CAVI in CKD stages 1 to 3a. Furthermore, based on the previous study, CAVI was decreased significantly in pioglitazone users but remained unchanged after treatment with sulfonylureas medicine like glimepiride [33].

The strength of our study is that this is the first and only study that compares the three measures of TBI, ABI, and CAVI together among patients with DKD. In the previous studies of Japan and the United States, only two measures with TBI and ABI were compared. The limitation of this study is that the gold standard for diagnosing PAD like angiography was not conducted in our study, because of the risk of progressing to hemodialysis after the procedures, especially for the advanced stages of CKD. The other limitation was that we were not able to include the smoking history of the study population.

In conclusion, TBI has a higher yield of PAD compared to ABI in this sample of Taiwanese patients with diabetic kidney disease, especially in advanced stages of CKD. CAVI may play a role in the early stage of DKD.

## Figures and Tables

**Table 1 jcm-12-07393-t001:** (**A**) Distribution of subjects according to clinical characteristics and ankle-branchial index (ABI) results. (**B**) Distribution of subjects according to clinical characteristics and toe-branchial index (TBI) results. (**C**) Distribution of subjects according to clinical characteristics and cardio-ankle vascular index (CAVI) results.

**(A)**				
		**ABI**		***p*-Value**
	**All Patients** **N = 368**	**Abnormal (<0.9)** **N = 32 (8.7%)**	**Normal (>0.9)** **N = 336 (91.3%)**	
Normal (reference)	82	3 (9.4%)	79 (23.5%)	0.0067 *
Chronic Kidney Disease				
Stage 1	39	3 (9.4%)	36 (10.7%)	
Stage 2	60	4 (12.5%)	56 (16.7%)	.
Stage 3a	92	5 (15.6%)	87 (25.9%)	.
Stage 3b	60	9 (28.1%)	51 (15.2%)	.
Stage 4 + 5	35	8 (25.0%)	27 (8.0%)	.
Age (years)	368	68.59 ± 13.14	67.03 ± 10.76	0.4411
Diabetes mellitus duration (years)	368	18.07 ± 9.11	15.60 ± 9.46	0.1711
Sex				
Male	190	13 (40.6%)	177 (52.7%)	0.1629
Female	178	19 (59.4%)	159 (47.3%)	.
Urine ACR (mg/g)				
0–29	159	10 (31.3%)	149 (44.3%)	0.0396 *
30–299	137	10 (31.3%)	127 (37.8%)	
≥300	72	12 (37.5%)	60 (17.9%)	
Total Cholesterol (mg/dL)	290	151.20 ± 25.43	153.30 ± 37.73	0.7184
Triglycerides (mg/dL)	366	133.30 ± 80.11	119.20 ± 92.55	0.4137
High-density lipoprotein (mg/dL)	365	46.18 ± 16.94	49.07 ± 15.66	0.3306
Low-density lipoprotein (mg/dL)	368	69.07 ± 20.67	75.28 ± 26.67	0.2015
Creatinine (mg/dL)	365	1.79 ± 1.49	1.35 ± 1.38	0.0843
Fasting glucose (mg/dL)	360	143.70 ± 54.81	133.30 ± 39.07	0.3055
Hemoglobin A1C (%)	368	7.09 ± 1.23	7.43 ± 4.60	0.3063
Body mass index (kg/m^2^)	367	27.64 ± 5.68	26.92 ± 5.59	0.4866
Systolic blood pressure (mmHg)	367	137.20 ± 16.92	131.50 ± 15.28	0.0487 *
Diastolic blood pressure (mmHg)	367	75.88 ± 10.52	75.14 ± 11.26	0.7232
Hypertension	260	24 (75.0%)	236 (70.2%)	0.5719
Dyslipidemia	306	27 (84.4%)	279 (83.0%)	0.8466
Cardiovascular Disease	88	14 (43.8%)	74 (22.0%)	0.0059 *
Antidiabetic agent				
Sulfonylureas	182	15 (46.9%)	167 (49.7%)	0.7599
Metformin	257	16 (50.0%)	241 (71.7%)	0.0105 *
Meglitinides	15	3 (9.4%)	12 (3.6%)	0.1126
Thiazolidinediones	123	8 (25.0%)	115 (34.2%)	0.2904
Alpha-glucosidase inhibitors	31	3 (9.4%)	28 (8.3%)	0.8394
Dipeptidyl-peptidase 4 inhibitors	124	7 (21.9%)	117 (34.8%)	0.1387
Sodium-glucose Co-transporter 2 inhibitors	152	12 (37.5%)	140 (41.7%)	0.6474
Insulin	114	14 (43.8%)	100 (29.8%)	0.1020
Antihypertensives				
Angiotensin converting enzyme inhibitors	19	0 (0%)	19 (5.7%)	0.1665
Angiotensin II receptor blocker	175	17 (53.1%)	158 (47.2%)	0.5189
Beta-blocker	93	9 (28.1%)	84 (25.1%)	0.7047
Calcium Channel Blocker	166	17 (53.1%)	149 (44.5%)	0.3477
Diuretic	40	3 (9.4%)	37 (11.0%)	0.7721
Alpha-blocker	21	4 (12.5%)	17 (5.1%)	0.0840
Antihyperlipidemic agent				
Stain	292	27 (84.4%)	265 (78.9%)	0.4622
Fibrates	40	3 (9.4%)	37 (11.0%)	0.7762
Other lipid-lowering agents	25	1 (3.1%)	24 (7.1%)	0.3881
(**B**)				
		**TBI**		***p*-Value**
	**All Patients** **N = 368**	**Abnormal (<0.65)** **N = 78 (21.2%)**	**Normal (>0.65)** **N = 290 (78.8%)**	
Normal (reference)	82	7 (9.4%)	75 (23.5%)	0.0039 *
Chronic Kidney Disease				
Stage 1	39	6 (7.7%)	33 (11.4%)	
Stage 2	60	11 (14.1%)	49 (16.9%)	.
Stage 3a	92	25 (32.1%)	67 (23.1%)	.
Stage 3b	60	16 (20.5%)	44 (15.2%)	.
Stage 4 + 5	35	13 (16.7%)	22 (7.6%)	.
Age (years)	368	69.09 ± 11.03	66.64 ± 10.92	0.0808
Diabetes mellitus duration (years)	368	17.37 ± 9.27	15.38 ± 9.46	0.1057
Sex				
Male	190	44 (56.4%)	146 (50.3%)	0.3413
Female	178	34 (43.6%)	144 (49.7%)	.
Urine ACR (mg/g)				
0–29	159	30 (38.5%)	129 (44.5%)	0.5543
30–299	137	30 (38.5%)	107 (36.9%)	
≥300	72	18 (23.1%)	54 (18.6%)	
Total Cholesterol (mg/dL)	290	151.70 ± 35.65	153.50 ± 37.26	0.7273
Triglycerides (mg/dL)	366	111.90 ± 61.97	122.70 ± 97.99	0.2366
High-density lipoprotein (mg/dL)	365	47.60 ± 14.28	49.15 ± 16.16	0.4408
Low-density lipoprotein (mg/dL)	368	74.51 ± 25.91	74.80 ± 26.37	0.9310
Creatinine (mg/dL)	365	1.62 ± 1.71	1.32 ± 1.29	0.1540
Fasting glucose (mg/dL)	360	141.10 ± 55.19	132.30 ± 35.80	0.1927
Hemoglobin A1C (%)	368	7.24 ± 1.38	7.44 ± 4.92	0.5378
Body mass index (kg/m^2^)	367	26.80 ± 5.80	27.03 ± 5.55	0.7496
Systolic blood pressure (mmHg)	367	132.10 ± 18.38	132.01 ± 14.64	0.9871
Diastolic blood pressure (mmHg)	367	74.56 ± 12.19	75.38 ± 10.92	0.5697
Hypertension	260	58 (74.4%)	202 (69.7%)	0.4180
Dyslipidemia	306	68 (87.2%)	238 (82.1%)	0.2844
Cardiovascular Disease	88	31 (39.7%)	57 (19.7%)	0.0002 *
Antidiabetic agent				
Sulfonylureas	182	37 (47.4%)	145 (50.0%)	0.6876
Metformin	257	48 (61.5%)	209 (72.1%)	0.0720
Meglitinides	15	8 (10.3%)	7 (2.4%)	0.0019 *
Thiazolidinediones	123	31 (39.7%)	92 (31.7%)	0.1826
Alpha-glucosidase inhibitors	31	10 (12.8%)	21 (7.2%)	0.1153
Dipeptidyl-peptidase 4 inhibitors	124	30 (38.5%)	94 (32.4%)	0.3158
Sodium-glucose Co-transporter 2 inhibitors	152	28 (35.9%)	124 (42.8%)	0.2746
Insulin	114	25 (32.1%)	89 (30.7%)	0.8174
Antihypertensives				
Angiotensin converting enzyme inhibitors	19	3 (3.8%)	16 (5.5%)	0.5499
Angiotensin II receptor blocker	175	41 (52.6%)	134 (46.4%)	0.3308
Beta-blocker	93	20 (25.6%)	73 (25.3%)	0.9452
Calcium Channel Blocker	166	35 (44.9%)	131 (45.3%)	0.9426
Diuretic	40	18 (23.1%)	22 (7.6%)	0.0001 *
Alpha-blocker	21	4 (5.1%)	17 (5.9%)	0.7991
Antihyperlipidemic agent				
Stain	292	66 (84.6%)	226 (77.9%)	0.1955
Fibrates	40	8 (10.3%)	32 (11.0%)	0.8446
Other lipid-lowering agents	25	9 (11.5%)	16 (5.5%)	0.0607
(**C**)				
		**CAVI**		***p*-Value**
	**All Patients** **N = 368**	**Abnormal (>9)** **N = 168 (45.7%)**	**Normal (<9)** **N = 200 (54.3%)**	
Normal (reference)	82	41 (50.0%)	41 (50.0%)	0.0820
Chronic Kidney Disease				
Stage 1	39	11 (6.5%)	28 (14.0%)	
Stage 2	60	22 (13.1%)	38 (19.0%)	.
Stage 3a	92	44 (26.2%)	48 (24.0%)	.
Stage 3b	60	33 (19.6%)	27 (13.5%)	.
Stage 4 + 5	35	17 (10.1%)	18 (9.0%)	.
Age (years)	368	71.91 ± 8.64	63.18 ± 11.16	<0.0001 *
Diabetes mellitus duration (years)	368	17.77 ± 9.31	14.13 ± 9.25	0.0003 *
Sex				
Male	190	91 (54.2%)	99 (49.5%)	0.3722
Female	178	77 (45.8%)	101 (50.5%)	.
Urine ACR (mg/g)				
0–29	159	78 (46.4%)	81 (40.5%)	0.5200
30–299	137	59 (35.1%)	78 (39.0%)	
≥300	72	31 (18.5%)	41 (20.5%)	
Total Cholesterol (mg/dL)	290	151.90 ± 36.02	154.20 ± 37.66	0.5998
Triglycerides (mg/dL)	366	121.20 ± 89.95	119.70 ± 93.10	0.8740
High-density lipoprotein (mg/dL)	365	47.53 ± 14.25	49.92 ± 16.92	0.1422
Low-density lipoprotein (mg/dL)	368	73.64 ± 24.57	75.65 ± 27.59	0.4656
Creatinine (mg/dL)	365	1.43 ± 1.51	1.35 ± 1.29	0.5667
Fasting glucose (mg/dL)	360	135.90 ± 41.08	132.70 ± 40.37	0.4479
Hemoglobin A1C (%)	368	7.55 ± 4.98	7.27 ± 3.87	0.5531
Body mass index (kg/m^2^)	367	25.69 ± 5.20	28.06 ± 5.70	<0.0001 *
Systolic blood pressure (mmHg)	367	132.70 ± 15.17	131.50 ± 5.76	0.4610
Diastolic blood pressure (mmHg)	367	75.17 ± 11.67	75.23 ± 10.80	0.9618
Hypertension	260	120 (71.4%)	140 (70.0%)	0.7643
Dyslipidemia	306	136 (81.0%)	170 (85.0%)	0.3015
Cardiovascular Disease	88	34 (20.2%)	54 (27.0%)	0.1298
Antidiabetic agent				
Sulfonylureas	182	82 (48.8%)	100 (50.0%)	0.8200
Metformin	257	110 (65.5%)	147 (73.5%)	0.0948
Meglitinides	15	12 (7.1%)	3 (1.5%)	0.0064 *
Thiazolidinediones	123	56 (33.3%)	67 (33.5%)	0.9731
Alpha-glucosidase inhibitors	31	20 (11.9%)	11 (5.5%)	0.0276 *
Dipeptidyl-peptidase4 inhibitors	124	62 (36.9%)	62 (31.0%)	0.2326
Sodium-glucose Co-transporter 2 inhibitors	152	69 (41.1%)	83 (41.5%)	0.9337
Insulin	114	59 (35.1%)	55 (27.5%)	0.1154
Antihypertensives				
Angiotensin converting enzyme inhibitors	19	7 (4.2%)	12 (6.0%)	0.4221
Angiotensin II receptor blocker	175	81 (48.2%)	94 (47.2%)	0.8517
Beta-blocker	93	39 (23.2%)	54 (27.1%)	0.3895
Calcium Channel Blocker	166	78 (46.4%)	88 (44.2%)	0.6721
Diuretic	40	18 (10.7%)	22 (11.1%)	0.9168
Alpha-blocker	21	5 (3.0%)	16 (8.0%)	0.0374 *
Antihyperlipidemic agent				
Stain	292	130 (77.4%)	162 (81.0%)	0.3930
Fibrates	40	18 (10.7%)	22 (11.0%)	0.9301
Other lipid-lowering agents	25	12 (7.1%)	13 (6.5%)	0.8071

* *p*-value < 0.05.

**Table 2 jcm-12-07393-t002:** Multivariate analysis of ABI, TBI, and CAVI with different parameters.

	ABI			TBI			CAVI		
	AOR	95% CI	*p*-Value	AOR	95% CI	*p*-Value	AOR	95% CI	*p*-Value
Chronic kidney disease									
Stage 1	16.00	0.58–443.12	0.1018	6.17	0.89–42.61	0.0651	0.16	0.03–0.77.	0.0223 *
Stage 2	4.93	0.22–112.26	0.3173	5.49	0.98–30.87	0.0535	0.18	0.04–0.74.	0.0180 *
Stage 3a	1.30	0.10–17.22	0.8400	6.50	1.63–25.97	0.0080 *	0.31	0.10–0.93.	0.0375 *
Stage 3b	6.54	0.47–91.56	0.1631	7.47	1.52–36.81	0.0135 *	0.38	0.10–1.44.	0.1552
Stage 4 + 5	11.34	0.22–589.91	0.2285	20.13	1.96–206.92	0.0116 *	0.35	0.04–3.03.	0.3388
Age	1.05	0.98–1.13	0.1747	1.00	0.96–1.05	0.8878	1.11	1.07–1.16.	<0.0001 *
Diabetes mellitus	0.98	0.91–1.06	0.6270	1.00	0.96–1.04	0.9383	1.05	1.01–1.09.	0.0282 *
duration (years)									
Sex	1.11	0.28–4.37	0.8778	1.62	0.75–3.53	0.2218	1.11	0.57–2.16.	0.7560
Urine ACR (mg/g) 30–299	0.15	0.02–1.47	0.1022	0.78	0.24–2.54	0.6820	1.45	0.49–4.34.	0.5042
≥300	0.18	0.01–2.71	0.2157	0.57	0.14–2.36	0.4410	2.20	0.54–8.99.	0.2713
Total Cholesterol (mg/dL)	0.99	0.91–1.07	0.8015	1.00	0.94–1.06	0.9974	0.95	0.91–1.00.	0.0556
Triglycerides (mg/dL)	1.01	0.99–1.03	0.5685	1.00	0.98–1.01	0.4962	1.01	1.00–1.02.	0.0120 *
High-density lipoprotein (mg/dL)	1.01	0.91–1.13	0.7916	0.99	0.92–1.06	0.7823	1.04	0.98–1.10.	0.2218
Low-density lipoprotein (mg/dL)	1.01	0.92–1.11	0.7942	1.01	0.94–1.07	0.8820	1.06	1.00–1.12.	0.0388 *
Creatinine (mg/dL)	0.81	0.36–1.78	0.5926	0.96	0.73–1.28	0.7893	0.80	0.60–1.07.	0.1350
Fasting glucose (mg/dL)	1.01	0.99–1.04	0.2031	1.01	1.00–1.02	0.1551	1.00	0.99–1.01.	0.5792
Hemoglobin A1C (%)	0.65	0.29–1.43	0.2842	0.96	0.81–1.14	0.6725	1.06	0.97–1.16.	0.1886
Body mass index (kg/m^2^)	1.01	0.88–1.17	0.8839	0.96	0.90–1.03	0.2220	0.92	0.85–0.99.	0.0223 *
Systolic blood pressure (mmHg)	1.12	1.03–1.21	0.0090	1.01	0.97–1.06	0.5000	1.00	0.96–1.03.	0.7998
Diastolic blood pressure (mmHg)	0.91	0.82–1.01	0.0612	1.00	0.95–1.05	0.9161	1.01	0.96–1.06.	0.6553
Hypertension	0.52	0.05–5.87	0.5960	1.26	0.34–4.61	0.7274	0.93	0.29–2.92.	0.8971
Dyslipidemia	-	-	-	0.82	0.06–11.86	0.8843	0.52	0.06–4.31.	0.5415
Cardiovascular Disease	7.19	1.75–29.59	0.0063 *	4.00	1.76–9.11	0.0010 *	0.50	0.23–1.06.	0.0711
Antidiabetic agent									
Sulfonylureas	1.53	0.31–7.56	0.6023	0.88	0.39–2.01	0.7590	1.01	0.49–2.08.	0.9884
Metformin	0.75	0.12–4.84	0.7631	2.53	0.84–7.64	0.0996	0.93	0.36–2.37.	0.8702
Meglitinides	7.84	0.23–265.48	0.2518	6.13	1.05–35.87	0.0444 *	47.27	2.63–848.93.	0.0089 *
Thiazolidinediones	0.92	0.18–4.66	0.9200	2.14	0.87–5.26	0.0984	0.90	0.40–2.01.	0.7971
Alpha-glucosidase inhibitors	0.45	0.04–4.75	0.5086	1.14	0.32–4.09	0.8447	2.55	0.77–8.45.	0.1255
Dipeptidyl-peptidase 4 inhibitors	0.10	0.01–0.67	0.0177*	0.41	0.15–1.17	0.0974	0.93	0.37–2.33.	0.8816
Sodium-glucose Co-transporter 2 inhibitors	0.89	0.17–4.67	0.8924	0.55	0.20–1.47	0.2309	1.71	0.70–4.18.	0.2361
Insulin	1.31	0.22–7.95	0.7713	0.51	0.20–1.33	0.1699	1.01	0.46–2.22.	0.9812
Antihypertensives									
Angiotensin converting enzyme inhibitors	1.51	0.41–11.15	0.7230	1.76	0.29–10.61	0.5367	0.55	0.11–2.78.	0.4661
Angiotensin II receptor blocker	1.49	0.22–10.04	0.6830	1.35	0.46–3.93	0.5809	0.69	0.26–1.80.	0.4442
Beta-blocker	1.19	0.21–6.66	0.8469	1.09	0.45–2.67	0.8438	1.16	0.52–2.60.	0.7202
Calcium Channel Blocker	0.64	0.12–3.61	0.6173	0.45	0.19–1.06	0.0662	1.64	0.73–3.69.	0.2283
Diuretics	0.60	0.05–7.65	0.6958	4.19	1.36–12.90	0.0125 *	1.39	0.46–4.16.	0.5615
Alpha-blocker	6.61	0.66–65.85	0.1072	0.55	0.12–2.60	0.4486	0.13	0.03–0.60.	0.0088 *
Antihyperlipidemic agent									0.7230
Stain	0.61	0.10–8.41	0.7585	1.65	0.15–18.70	0.6863	1.42	0.21–9.72.	
Fibrates	0.56	0.04–7.46	0.6586	1.23	0.34–4.50	0.7528	2.90	0.80–10.55.	0.1052
Other lipid-lowering agent	0.81	0.15–9.81	0.8791	1.85	0.44–7.75	0.4027	0.56	0.16–1.97.	0.3655

* *p*-value < 0.05; adjusted odds ratio, AOR.

## Data Availability

The data presented in this study are available on request from the corresponding author. The data are not publicly available due to privacy.

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
