# Peer review of "Better Detection of Peripheral Arterial Disease with Toe-Brachial Index Compared to Ankle-Brachial Index among Taiwanese Patients with Diabetic Kidney Disease"

_jcm, 2023, doi:10.3390/jcm12237393_

Round 1

Reviewer 1 Report

Comments and Suggestions for Authors

There are several recommendations to improve the paper which are in the attachment. 

Comments on the Quality of English Language

Some of the sentences are extremely long and may have to be rewritten to improve readability. 

Author Response

  1. Title: The title is unclear – what exactly are being associated with each other?

What are the dependent and independent variables?

Maybe what is meant is this- “Toe-brachial index is more strongly associated with

severity of chronic kidney disease (or CKD stages) than ankle-brachial index among

Taiwanese patients” OR

“Toe-brachial index Detected Peripheral Arterial Disease in Higher CKD Stages

Better than Ankle-Brachial Index among Taiwanese Patients”

“Better Detection of Peripheral Arterial Disease with Toe-Brachial Index compared

to Ankle-Brachial Index among Taiwanese Patients with Diabetic Kidney Disease”

Response: Thank you for your suggestion. We chose the third one as our title.

  1. Abstract
  2. Please use a structured abstract: Background/Objectives, Materials and

Methods, Results, Conclusion

Response: As per your suggestion, we use structured abstract now.

  1. The aim is unclear; from the analysis/results it appears that the authors

wanted to determine which among the measures will have the highest yield

for PAD among patients with CKD

Response: We revised our statement. Please reconsider.

  1. The results that are given describes the confidence interval but not the

point estimate. And what is this statistic? is this relative risk or odds ratio

for PAD?

Response: Multiple forward logistic regression analysis was used in the analysis. The detailed description was provided now in methods and material. The odds ratio for PAD was added now in the abstract.

  1. What is meant by “TBI is more significant than ABI in evaluating PAD”?

Given that this is a cross-sectional study, does this mean that the estimated

odds of having PAD using TBI versus ABI is higher for the former.

Response: Yes. To make it clear, we revised our statement. Please reconsider.

  1. On the other hand, for CAVI, the estimated OR is less than 1 which means

that the ABI seems to be better. Please clarify.

Response: Yes. We concluded that ABI is better than CAVI in diagnosing PAD in DKD.

  1. The conclusion is also wrong. It says: “In conclusion, TBI is associated more

strongly with ABI in diabetic kidney diseases in Taiwanese patients in diagnosing PAD.”

There is no attempt to determine the association of TBI to ABI. The objective is to

determine which of the tests is more strongly associated with the diagnosis of PAD

among patients with chronic kidney disease. So the more appropriate conclusion is

that there is better yield of detection of peripheral arterial disease with TBI

compared to ABI among Taiwanese Patients with Diabetic Kidney Disease.

Response: Thank you for your comments. We rephrased it as your comments.

  1. Introduction
  2. Please improve writing style. The first sentence in page 2, paragraph 1 is

too long spanning lines 48-52. Break up the sentence into 2.

Response: Thank you for your comments. We break up the sentences into 2 already on line 45-46.

  1. The rationale for doing the study is not clear. Have these 3 measures of

PAD already been compared in other populations (other nationalities)

especially among persons with diabetes and CKD?
Response:
In the previous studies of Japan and United States, only two measures with TBI and ABI were compared. Our study is the first and only on comparing three measures of TBI, ABI and CAVI together among patients with DKD. This statement was added to the discussion as the strength of our study.

  1. The objective of the study which is to investigate the potential association

of DKD patients to ABI, TBI, and CAVI is unclear since the greater majority of

patients have diabetic kidney disease since only 99 out of 368 had CKD

Stage 1-2. So maybe the better objective is to either determine the

association of the severity of CKD to the ABI, TBI and CAVI, OR to determine

the association of CKD (defined as eGFR less than 60 or CKD stage 3A

upwards) with ABI, TBI and CAVI. A better aim is actually to determine

which among the 3 measures (ABI, TBI and CAVI) has the highest detection

of PAD among patients with diabetic kidney disease.

Response: Thank you for nice comments. We rephrased our statement as your comments.

  1. Materials and Methods
  2. Describe the study site. Is it a general hospital or is it a specialty hospital for

persons with diabetes. How well does it represent the Taiwanese

population?

Response: This was well described at Materials and Methods study patients section.

  1. What is the study design? Calculated sample size ?

Response: The calculated sample size now was added now in statistical analysis.

  1. Who made each of the 3 measurements? Was a single person doing each

test across all the patients and if not, how was it ensured that the

measurements and readings were standardized?

Response: Yes, and its emphasized now in our Assessment of ABI, TBI and CAVI section.

  1. There is no section on how data was collected? Was a standard form used

for all patients?

Response: This was already well described in collection of demographic, medical, and laboratory data section.

  1. There is no section on statistical analysis or statistical design

Response: There is a statistical analysis section added now.

  1. Results
  2. Please improve the table titles. Table 1A is NOT “baseline” characteristics

since there was no second data collection. It is better labeled as

“Distribution of Subjects According to Clinical Characteristics and ABI

results”. Same is true for Table 1B and 1C. These are not “baseline”

characteristics.

Response: Thank you for your comments. Table titles were now revised according to your suggestions.

  1. If the intention is to compare which of the 3 measures have better yield in

CKD, then maybe the results for CKD 3A-CKD5 should have been combined

for Tables 1A to 1C.

Response: We initially intended to identify which stage of the CKD will start to yield a significant difference among the 3 measurements, therefore we break the CKD staging according to the KDIGO standard staging.

  1. In Table 1C, the % of normal and abnormal CAVI are incorrect. For the

abnormal AVI, N=168/368 (45.65% and not 8.7%) while for the normal AVI

N=200/368 is 54.35% and not 91.3%.

Response: Sorry for the mistakes. Now, it’s corrected.

  1. Stage 4-5 were combined in Table 2 since these are the higher severity CKD,

could you also have combined CKD Stage 3A-Stage 5 since these categories

comprise true CKD (eGFR less than 60)? What is AOR- approximate odds

ratio? Please put this at the bottom of the table. In the results, please state

that the AOR of TBI shows a value consistently above 1.0 beginning at CKD

3A, implying that this measure is associated with the true CKD stages.

Response: As the response above, we initially intended to identify which stage of the CKD will start to yield a significant difference among the 3 measurements, therefore we break the CKD staging according to the KDIGO standard staging. However, we also performed the statistic analysis according to your suggestion, and we obtained the same result as below. Therefore, we decided to keep the original staging of CKD.

AOR is adjusted odds ratio, and now it is put at the bottom of the table.

The statement of AOR of TBI was added now.

  1. Please summarize the key results of the study in the last paragraph of the

results. These findings are actually in the first paragraph of the discussion

but are never really emphasized in the results section.

Response: It’s added now at the end of the result section.

  1. Discussion
  2. The discussion includes relevant similar literature ( should we cut the reference?)

Response: Only reference 18-35 were left in the discussion, to make it easy for the readers to search for the references if publication is possible in the future, we prefer to keep the original references.

  1. Please include the limitations of your study either in the discussion or

conclusion e.g. the gold standard of angiography was not done.

Response: The strength and limitation of this study was now added at the end of the discussion.

  1. Conclusion
  2. Please correct the wording of the conclusion: “In conclusion, TBI has associated

more strongly over ABI in diabetic kidney diseases in Taiwanese patients in diagnosing

PAD. CAVI may play a role in the early stages of diabetic kidney disease and metabolic

syndrome.” This should instead be, “In conclusion, TBI has a higher yield of

PAD compared to ABI in this sample of Taiwanese patients with diabetic

kidney disease, especially at the higher CKD stages. CAVI may play a role in

the early stage of diabetic kidney disease.

Response: Thank you for your comments. We revised as your suggestion.

Reviewer 2 Report

Comments and Suggestions for Authors

Journal: JCM; Manuscript ID: jcm-2695261

Title: "Toe–brachial index is associated more strongly over ankle–brachial index in diabetic kidney disease Taiwanese patients"

Authors: Chia-Wei Chang et al.

One major and critical flaw in the paper is the absence of information about the statistical approach employed and the details regarding the data analysis methodology. This omission significantly restricts our ability to assess and evaluate the findings of the present study. Please consider adding a statistical analysis section. A few additional comments to be considered.

Comments:

1.     In the abstract conclusion, it would be beneficial to include a brief statement regarding the findings related to CAVI.

2.     The authors could consider improving clarity of the text: “Accordingly, CAVI is a newly developed method used to assess arterial stiffness such as common iliac, femoral, and tibial artery levels, in addition, these are not affected by blood pressure, and arterial stiffness is a predictor of CVD in CKD patients. Also, the study investigated CAVI to assess whether is a good tool to evaluate PAD in DKD patients is still under investigation.”. Likewise, it is difficult to interpret other parts of the text, e.g., “CKD was categorized into CKD stages 1 to 5. normal ABI(ABI>0.9 to<1.3) is more than in 133 abnormal ABI (ABI < 0.9 or ≧ 1.3) with a p-value of 0.0067.” It is imperative that the authors meticulously review the manuscript for linguistic clarity and consider potential enhancements to improve its overall readability.

3.     The authors should consider rewriting the manuscript to enhance its tone and simplicity while maintaining formality and robustness. For instance, in the sentence: “The association between CAVI and age was linear [32], the older the accurate are, 261 the higher level they have.” the authors might consider describing this type of association as a positive one.

4.     Please include a paragraph discussing the limitations of the study. Additionally, provide a summary of the study's strengths and the clinical implications of the findings.

5.     In the authors' statement: “In our study, patients who used non-sulfonylureas meglitinides were associated with CAVI and TBI.” could the authors propose potential mechanisms or explanations for this finding? How do they interpret these results?

6.     Did the authors consider adjusting for smoking?

7.     It would be beneficial if the authors could clarify their methodology for evaluating the statement, “CAVI is also more significant over ABI”. Is their interpretation solely based on the p-value?

8.     Did the authors measure the indices for both sides of the limbs, i.e., the right and left sides? Further clarification on this point is needed.

9.     Please ensure that all abbreviations are defined the first time they are used in the abstract and main text (e.g., PAOD).

Comments on the Quality of English Language

Please refer to my relevant comments mentioned in the "Comments and Suggestions for Authors
" section. 

Author Response

One major and critical flaw in the paper is the absence of information about the statistical approach employed and the details regarding the data analysis methodology. This omission significantly restricts our ability to assess and evaluate the findings of the present study. Please consider adding a statistical analysis section. A few additional comments to be considered.

 Response: The statistical analysis section was now added at the end of the materials and methods section.

Comments:

  1. In the abstract conclusion, it would be beneficial to include a brief statement regarding the findings related to CAVI.

Response: Thank you. It’s added now.

  1. The authors could consider improving clarity of the text: “Accordingly, CAVI is a newly developed method used to assess arterial stiffness such as common iliac, femoral, and tibial artery levels, in addition, these are not affected by blood pressure, and arterial stiffness is a predictor of CVD in CKD patients. Also, the study investigated CAVI to assess whether is a good tool to evaluate PAD in DKD patients is still under investigation.”. Likewise, it is difficult to interpret other parts of the text, e.g., “CKD was categorized into CKD stages 1 to 5. normal ABI(ABI>0.9 to<1.3) is more than in 133 abnormal ABI (ABI < 0.9 or â‰§3) with a p-value of 0.0067.” It is imperative that the authors meticulously review the manuscript for linguistic clarity and consider potential enhancements to improve its overall readability.

Response: Thank you for your comments. We rephrase above statements already for clarity.

  1. The authors should consider rewriting the manuscript to enhance its tone and simplicity while maintaining formality and robustness. For instance, in the sentence: “The association between CAVI and age was linear [32], the older the accurate are, 261 the higher level they have.” the authors might consider describing this type of association as a positive one.

Response: Thank you, We already rephrase according to your suggestion,

  1. Please include a paragraph discussing the limitations of the study. Additionally, provide a summary of the study's strengths and the clinical implications of the findings.

 Response: The strength and limitation of the study were now added at the end of discussion section.

  1. In the authors' statement: “In our study, patients who used non-sulfonylureas meglitinides were associated with CAVI and TBI.” could the authors propose potential mechanisms or explanations for this finding? How do they interpret these results?

Response: This was explained further now in discussion section line 319-321.

  1.  Did the authors consider adjusting for smoking?

Response: We’re not able to identify the smoking history of the study population and this was added in our study limitation.

  1. It would be beneficial if the authors could clarify their methodology for evaluating the statement, “CAVI is also more significant over ABI”. Is their interpretation solely based on the p-value?

Response: Yes. We based it on p value<0.05 in our statistical analysis result.

  1. Did the authors measure the indices for both sides of the limbs, i.e., the right and left sides? Further clarification on this point is needed.

Response: Yes. This was well described in the Assessment of ABI, TBI, and CAVI of method section. The diagnosis of PAD was defined as ABI < 0.9 or ≧ 1.3 in either leg or TBI below 0.65.

  1. Please ensure that all abbreviations are defined the first time they are used in the abstract and main text (e.g., PAOD).

Response: Yes, it’s uniform to be PAD now.

Reviewer 3 Report

Comments and Suggestions for Authors

This is a cross-sectional study of the performances of several biomarkers of vascular elasticity in Tawanese patients with Type 2 diabetes. ABI, TBI, and CAVI were studied. The main conclusion of this study is that alterations in vascular elasticity are proportional to the severity of kidney disease. This is not a new finding. As this study is cross-sectional, comparisons of the predictive values of the various indexes were not possible.
I have the following comments :

Introduction : « Also, the study investigated CAVI to assess whether is a good tool to evaluate 60 PAD in DKD patients is still under investigation ». I can’t understand.

Definition of PAD, the studied outcome, is not a clinical one (« PAD was defined as ABI < 0.9 or 88 ≧ 1.3 in either leg or TBI below 0.65 »).

Lines 86-87, The ABI and TBI were calculated by the ratio of the ankle or toe systolic blood pressure divided by the arm systolic blood pressure. I can’t understand the diffence in calculation between ABI and TBI. Also, which arm, which ankle, ou toe ? Important issue to compare the performances of one against the pther.

Definition of PAD is not a clinical one : PAD was defined as ABI < 0.9 or 88 ≧ 1.3 in either leg or TBI below 0.65.

Abnormal ABI is a mix of patients with ABI<0.9 and of those with ABI>1.3. These patient categories are different. Please analyse seperately these two kinds of abnormality.

To calculate CAVI, blood viscosity and PWV must be measured. Please describe the methods used.

Tables 1 A,B,C : what is a « cardiovascular disease » ?

Table 2 : definition of AOR ?

Author Response

This is a cross-sectional study of the performances of several biomarkers of vascular elasticity in Tawanese patients with Type 2 diabetes. ABI, TBI, and CAVI were studied. The main conclusion of this study is that alterations in vascular elasticity are proportional to the severity of kidney disease. This is not a new finding. As this study is cross-sectional, comparisons of the predictive values of the various indexes were not possible.
I have the following comments :

Introduction : « Also, the study investigated CAVI to assess whether is a good tool to evaluate 60 PAD in DKD patients is still under investigation ». I can’t understand.

Response: CAVI is a new measurement that already substitute Pulse wave velocity (PWV) of the old machine. Therefore, the study of CAVI is still very few.

The definition of PAD, the studied outcome, is not a clinical one (« PAD was defined as ABI < 0.9 or 88 â‰§ 1.3 in either leg or TBI below 0.65 »).

Lines 86-87, The ABI and TBI were calculated by the ratio of the ankle or toe systolic blood pressure divided by the arm systolic blood pressure. I can’t understand the diffence in calculation between ABI and TBI. Also, which arm, which ankle, ou toe ? Important issue to compare the performances of one against the pther.

Response: This was well described in detail in our assessment of ABI , TBI and CAVI of Materials and Methods section.You can also refer to this website with youtube for how ABI and TBI were performed. https://hokansonvascular.com/articles/133517

Definition of PAD is not a clinical one : PAD was defined as ABI < 0.9 or 88 â‰§ 1.3 in either leg or TBI below 0.65.Abnormal ABI is a mix of patients with ABI<0.9 and of those with ABI>1.3. These patient categories are different. Please analyze seperately these two kinds of abnormality.

Response: We agree that the definition of abnormal ABI is either ABI <0.9 or >1.3. But the definition of the PAD is still with ABI <0.9 , and ABI >1.3 is for noncompressible arteries. In our study, we aimed to identify which measurement is better for the PAD patients in DKD population, therefore, we only included ABI <0.9 as our study population.

To calculate CAVI, blood viscosity and PWV must be measured. Please describe the methods used.

Response: The values of ABI, TBI, and CAVI will be determined from the measurements by VS-2000(Fukuda Denshi Co., Japan), which automatically and simultaneously measured blood pressures and pulse volumes in both arms and ankles or toe using an oscillometric method. Therefore, the blood viscosity is not needed now. The PWV was already substituted by the CAVI in the new generation of ABI measurements now.

Tables 1 A,B,C : what is a « cardiovascular disease » ?

Response: The cardiovascular diseases included histories of old MI, old stroke, atherosclerotic cardiovascular disease ,stable and unstable angina. And this was described now in Collection of Demographic, Medical, and Laboratory Data of Materials and Methods section.

Table 2 : definition of AOR ?

Response: Adjusted odds ratio ( AOR ) definition was put at bottom of the table 2 now.

Round 2

Reviewer 1 Report

Comments and Suggestions for Authors

Congratulations! The paper has been significantly improved and is now ready for manuscript editing! 

Reviewer 2 Report

Comments and Suggestions for Authors

Journal: JCM; Manuscript ID: jcm-2695261(Revised)

Title: "Toe–brachial index is associated more strongly over ankle–brachial index in diabetic kidney disease Taiwanese patients"

Authors: Chia-Wei Chang et al.

The authors have responded to my comments and suggestions and tried to make the necessary changes to the manuscript. The revised manuscript has been improved. There are no further comments. 

Reviewer 3 Report

Comments and Suggestions for Authors

The authors met my concerns